# Syringetin Promotes Melanogenesis in B16F10 Cells

**DOI:** 10.3390/ijms24129960

**Published:** 2023-06-09

**Authors:** Hyunju Han, Chang-Gu Hyun

**Affiliations:** Jeju Inside Agency and Cosmetic Science Center, Department of Chemistry and Cosmetics, Jeju National University, Jeju 63243, Republic of Korea; 00guswn00@naver.com

**Keywords:** B16F10, GSK3β/β-catenin, PI3K/Akt, MAPK, melanogenesis, PKA: syringetin

## Abstract

Syringetin, an active compound present in red grapes, jambolan fruits, *Lysimachia congestiflora*, and *Vaccinium ashei*, is a dimethyl myricetin derivative which contains free hydroxyl groups at the C-2′ and C-4′ positions in ring B. Recent studies have revealed that syringetin possesses multiple pharmacological properties, such as antitumor, hepatoprotective, antidiabetic, antioxidative, and cytoprotective activities. To date, there has been no attempt to test the action of syringetin on melanogenesis. In addition, the molecular mechanism for the melanogenic effects of syringetin remains largely unknown. In this study, we investigated the effect of syringetin on melanogenesis in a murine melanoma cell line from a C57BL/6J mouse, B16F10. Our results showed that syringetin markedly stimulated melanin production and tyrosinase activity in a concentration-dependent manner in B16F10 cells. We also found that syringetin increased MITF, tyrosinase, TRP-1, and TRP-2 protein expression. Moreover, syringetin inhibited ERK and PI3K/Akt phosphorylation by stimulating p38, JNK, PKA phosphorylation levels, subsequently stimulating MITF and TRP upregulation, resulting in the activation of melanin synthesis. Furthermore, we observed that syringetin activated phosphorylation of GSK3β and β-catenin and reduced the protein level of β-catenin, suggesting that syringetin stimulates melanogenesis through the GSK3β/β-catenin signal pathway. Finally, a primary skin irritation test was conducted on the upper backs of 31 healthy volunteers to determine the irritation or sensitization potential of syringetin for topical application. The results of the test indicated that syringetin did not cause any adverse effects on the skin. Taken together, our findings indicated that syringetin may be an effective pigmentation stimulator for use in cosmetics and in the medical treatment of hypopigmentation disorders.

## 1. Introduction

Hypopigmentation is a condition characterized by the appearance of white macules on the skin which can be diffused or localized, and may be acquired or congenital, with a specific distribution pattern. Acquired hypopigmentation disorders include vitiligo, tinea versicolor, pityriasis alba, and postinflammatory hypopigmentation, while congenital disorders include albinism, piebaldism, hypomelanosis of Ito, and tuberous sclerosis [1,2,3]. Hypopigmentation disorders can manifest due to various factors, such as a decrease in the number of melanocytes in the skin, impaired melanin production by melanocytes, or abnormal transfer of melanosomes to neighboring keratinocytes [4]. In the cosmeceutical field, melanogenesis modulators derived from natural ingredients are typically more attractive to customers. Currently, treatments for hypopigmentation include steroids, oral drugs, phototherapy, and transplants, but these methods raise concerns about safety and efficacy [5]. For example, topical corticosteroids used to treat vitiligo can cause skin atrophy, and 2.5% selenium sulfide, used to treat pityriasis versicolor, can cause contact dermatitis. Oral treatments such as imidazoles and triazoles can also cause gastrointestinal disturbances, such as nausea and vomiting [6,7,8,9,10]. As a result, there is a growing interest in developing new hypopigmentation treatments derived from natural sources, which are more appealing to customers and have fewer side effects.

Several intrinsic and extrinsic factors involved in mammalian skin pigmentation melanogenesis have been identified, including tyrosinase (TYR), tyrosinase-related protein 1 (TRP-1), and tyrosinase-related protein 2 (TRP-2 or dopachrome tautomerase) [11]. Tyrosinase, a pivotal rate-limiting enzyme, catalyzes the hydroxylation of tyrosine to L-3,4-dihydroxyphenylalanine (L-DOPA) and the oxidation of L-DOPA to L-dopaquinone in the first two steps of melanogenesis. L-dopaquinone can be converted to cysteinyl-DOPA, which is further oxidized to pheomelanin, or spontaneously reacted to produce 5,6-dihydroxyindole (DHI) and 5,6-dihydroxyindole-2-carboxylic acid (DHICA), via dopachrome. The production of DHICA is accelerated by TRP-2 or copper ions. Subsequently, TRP-1 oxidizes DHICA to indole-5,6-quinone-2-carboxylic acid, which is eventually converted into eumelanin [12]. Microphthalmia-associated transcription factor (MITF), a key activator of the tyrosinase promoter, is a master transcription factor that activates the expression of melanogenic genes and is required for melanocyte differentiation, proliferation, and survival [13]. Therefore, stimulation of the activities of an enzymatic cascade consisting of several key enzymes and regulators, such as MITF, tyrosinase, TRP-1, and TRP-2, is considered an important strategy in the development of potent melanogenesis stimulators or dermatological drugs [14,15,16]. 

Numerous signaling pathways are involved in melanogenesis, including cAMP-dependent protein kinase A (PKA) and cAMP response element-binding protein (CREB), which stimulate the expression of MITF and lead to an increase in melanin levels [17,18]. The mitogen-activated protein kinase (MAPK) family proteins ERK, JNK, and p38 MAPK also play crucial roles in melanogenesis. Phosphorylation of p38 and JNK can induce MITF expression, whereas phosphorylation of ERK inhibits melanin synthesis [19,20]. Additionally, the PI3K/Akt signaling pathway has been identified as regulating melanogenesis through MITF expression and subsequent expression of tyrosinase, TRP-1, and TRP-2 [21,22]. The Wnt-signaling pathways are also important in melanogenesis, with β-catenin playing a key role in enhancing the expression of MITF and increasing melanin production [23,24]. Based on these findings, these signaling pathways have the potential to be developed as strategic targets for controlling melanin synthesis. 

Syringetin (3,4′,5,7-tetrahydroxy-3′,5′-dimethoxyflavone), an O-methylated flavonol belonging to the group of flavonoids, is present in grapes, jambolan fruits, Lysimachia congestiflora, and Vaccinium ashei. It has several pharmacological properties, including antidiabetic, antitumor, antiamnestic, bone-homeostasis promoting, and neuroprotective activities [25,26,27,28,29]. In addition, Bando et al. demonstrated that syringetin enhances radiosensitivity more effectively in cancer cells than in normal cells through enhancement of the caspase-3-mediated apoptosis pathway. It could be useful in the development of novel efficacious radiosensitizers [30]. However, to date, no study has focused on the effects of syringetin on melanogenesis. In this present study, we aimed to systematically investigate the effects of syringetin on melanin synthesis and unveil its underlying molecular mechanisms. Our results demonstrated that syringetin can activate melanogenesis in vitro by upregulating MITF and melanogenic protein expression through mediating the PKA/CREB, MAPK, AKT/PI3K, and GSK-β/Wnt/β-catenin signaling pathways. Therefore, we suggest that syringetin may have potential as a melanogenic stimulator for preventing hypopigmentation disorders in the future.

## 2. Results 

### 2.1. Syringetin Increased Melanin Content and Tyrosinase Activity in B16F10 Cells

To ensure that the effect of syringetin on melanin synthesis is not caused by cytotoxicity, we conducted an MTT assay to evaluate its cytotoxicity against B16 cells. We treated the cells with various concentrations of syringetin for 72 h and determined the intracellular melanin amount/protein amount and cell proliferation as percentage values. As shown in Figure 1, syringetin exhibited concentration-dependent melanogenesis-stimulation activity without any significant effect on cell proliferation at concentrations of 2–6 μg/mL, although 12.5 μg/mL of syringetin slightly decreased cell proliferation. The melanogenic activity of 1.5 μg/mL syringetin (86.7 nM) was about 1.6-fold of that of a reference compound, α-MSH, at a concentration of 100 nM. To further evaluate the effect of syringetin on melanin synthesis, we exposed the cells to syringetin (0.75–6 μg/mL) or α-MSH (100 nM), a positive control. The tyrosinase activity of the syringetin-treated cells was significantly increased compared to the level of the untreated control cells, indicating that the stimulative activity of syringetin on melanogenesis is due to an upregulation in tyrosinase activity.

### 2.2. Syringetin Regulated the Expression of Melanogenesis-Related Proteins in B16F10 Cells

Tyrosine was utilized as a substrate in the process of melanogenesis, wherein a sequence of enzymatic conversions facilitated by tyrosinase, TRP-1, and TRP-2 leads to the synthesis of eumelanin and pheomelanin. Given the demonstrated ability of syringetin to increase melanin and tyrosinase activity, further experiments were conducted to investigate the underlying mechanism. Specifically, we evaluated the impact of syringetin on three enzymes (TRP-1, TRP-2, and tyrosinase) involved in the melanogenic pathway. The results indicated that syringetin exhibited a dose-dependent induction in the expression of all three enzymes (Figure 2a–d). Additionally, we examined the effect of syringetin on MITF expression, a melanocyte-specific transcription factor that regulates the expression of these enzymes (as depicted in Figure 2e,f). The findings confirmed that syringetin successfully activates MITF expression in B16F10 cells. It can be inferred from the results that syringetin causes an increase in the expression of tyrosinase, TRP-1, and TRP-2 through the mediation of MITF, which leads to melanogenesis.

### 2.3. Syringetin Induced Melanogenesis in B16F10 Cells through the PKA Signaling Pathway

The PKA/MITF signaling cascade was examined using Western blotting to clarify how syringetin activated the expression of MITF and subsequently increased the expression levels of TRY and TRPs. cAMP is a crucial signal transduction molecule that activates the phosphorylation of PKA, which is one of the primary pathways that stimulates melanogenesis in response to extracellular stimuli. PKA activation then initiates two distinct signaling pathways that upregulate MITF expression: CREB and GSK3 [17,18]. Therefore, to investigate the transcription factor that stimulates MITF expression, we examined the phosphorylation levels of PKA, which is the downstream activator of cAMP signaling. Figure 3 shows that syringetin treatment significantly upregulated the expression level of phosphorylated PKA in comparison to the control treatment. As a result, the PKA/CREB signaling pathway was confirmed to be closely linked to the melanogenic effect of syringetin.

### 2.4. Syringetin Induced Melanogenesis in B16F10 Cells through the PI3K/Akt Signaling Pathways

The activation of MITF and subsequent melanin synthesis could be achieved by inhibiting the phosphorylation of AKT [21,22]. To investigate the effect of syringetin on melanogenesis, the modulation of AKT was studied. It was observed that syringetin treatment led to a concentration-dependent increase in p-AKT expression, indicating its melanogenic effect. The results suggested that syringetin upregulated the MITF/tyrosinase pathway by suppressing the AKT pathway.

### 2.5. Syringetin Induced Melanogenesis in B16F10 Cells through the MAPK Signaling Pathway

The MAPK family, including ERK, JNK, and p38 MAPK, has been shown to play a crucial role in melanogenesis. Although ERK and JNK have been implicated in regulating melanin production, the overall role of MAPK pathway activation in melanin synthesis remains controversial [19,20]. In B16 melanoma cells, theophylline enhances melanin production by increasing ERK phosphorylation, whereas it has been demonstrated that increasing p-ERK levels suppresses melanogenesis [23]. This complexity in the regulation of melanin synthesis may be due to the fact that phosphorylation increases MITF transcriptional activity while also inducing ubiquitin proteasome-dependent degradation of MITF. To understand the mechanism underlying syringetin’s effect on MITF-mediated melanogenesis, the phosphorylation of p38, ERK, and JNK was evaluated. The results showed that syringetin treatment led to a significant increase in ERK phosphorylation (3 μg/mL and 6 μg/mL) and a remarkable reduction in p38 and JNK phosphorylation, without affecting their total protein expression. These findings suggested that the melanogenic effect of syringetin may be partly related to the activation of ERK and the inhibition of p38 and JNK in melanogenesis.

### 2.6. Syringetin Repressed Melanogenesis in B16F10 Cells through the GSK-3β/β-Catenin Signaling Pathway

The Wnt/β-catenin signaling pathway is composed of the GSK-3 and β-catenin proteins. GSK-3 is a constitutively active kinase that can be phosphorylated by several kinases, including Akt and PKA. Recent studies have highlighted the close relationship between the β-catenin signaling pathway and melanin synthesis. β-catenin promotes MITF transcription by translocating from the cytoplasm to the nucleus, thereby upregulating MITF expression and binding to lymphocyte enhancing factor [23,24]. To investigate the role of Wnt/β-catenin pathways in syringetin’s stimulating effects on melanin production, we examined the expression of GSK-3β and β-catenin, two key molecules in the Wnt-signaling pathway. Our results showed that syringetin induced the expression of GSK3β and β-catenin in a concentration-dependent manner. Notably, syringetin treatment significantly reduced the phosphorylation of β-catenin. However, we did not observe significant changes in the levels of GSK3β and β-catenin in the cytoplasm. These findings suggest that the syringetin-mediated increase in melanin production may be triggered through the Wnt pathway. This is consistent with previous reports demonstrating that the activation of GSK-3/β-catenin signaling influences melanin production in B16F10 melanoma cells.

### 2.7. Syringetin Is Safe for Human Skin

In order to assess the suitability of syringetin for topical use, we conducted primary skin irritation tests on human volunteers. Using skin patches, we administered 3 μg/mL and 6 μg/mL concentrations of syringetin and monitored the patches 20 min and 24 h after the removal of the substance. To serve as a control, we utilized squalene (a solvent). Our findings, as displayed in Table 1, demonstrated that syringetin caused “no to slight irritation” as a primary irritant to human skin. 

## 3. Discussion

Flavonoids encompass various significant compound groups, characterized by a common core backbone structure known as flavan. These groups consist of flavones, flavonols (3-hydroxyflavone), flavanols, isoflavones, and anthocyanidins, each exhibiting distinct modifications in their side groups. The fundamental structure of flavan consists of three rings: A, B, and C. The differentiation among the aforementioned major groups of flavonoids arises from specific alterations in the side group modifications on these rings. These side groups play pivotal roles in determining the activities and functions of these compounds, as even slight modifications can yield significantly diverse effects. Due to their beneficial bioactivities, including antioxidative, anti-inflammatory, and anticancer properties, along with their low toxicity, flavonoids have emerged as highly valuable natural resources in the field of healthcare. They find widespread application in supplemental foods and cosmetics [31,32]. 

Numerous studies have demonstrated that flavonoids play a regulatory role in melanogenesis by influencing melanogenic proteins, particularly tyrosinase. Interestingly, despite their structural similarities, different flavonoids exhibit contrasting effects on melanin synthesis, highlighting the intriguing and complex nature of their actions in this process [33,34,35]. Conflicting results have been reported regarding the association between the hydroxy group of flavonoids and their structure and activity against melanogenesis. Takekoshi et al. [36] reported that quercetin, kaempferol, and rhamnetin, which possess only hydroxyl groups at the 4’ position of the flavonoid skeleton ring B, have melanogenesis-stimulating effects in human melanoma HMV II cells. However, the presence of additional hydroxyl groups, such as robinetin and myricetin, in addition to the 4′ position of ring B did not affect the activation of melanogenesis. On the other hand, Horibe et al. [37] reported that luteolin and kaempferol had no effect on melanogenesis in B16F10 cells, but 4′-O-methylated flavonoids, diosmetin (4′-O-methylluteolin), acacetin (4′-O-methylapigenin), or kaempferide (4′-O-methylkaempferol) increased the melanin contents of the cells 3- to 7-fold higher than the control cells. Furthermore, Nakashima et al. [38] reported that luteolin and quercetin actually inhibited melanin production in B16 melanoma 4A5. Although these effects are controversial, the structure–activity relationship (SAR) of flavonoids is not clear. Therefore, we attempted to investigate how the functionalities of flavonoids affect melanin biosynthesis. We focused on the effect of the polymethoxy group on the B-ring of flavonoids because our previous study, which compared the effects of chrysoeriol and its derivatives including luteolin and apigenin, suggested that the most effective melanin stimulation was observed when a hydroxyl group was substituted at C-4 in the B-ring of flavone, while a reversible decrease in melanin-stimulating activity was observed with a hydroxyl group substitution at C-3. Additionally, the comparison of chrysoeriol and luteolin in terms of SAR confirmed that replacing the hydroxyl group at the C-3 position with a methoxy group restored the activation of melanogenesis. We selected syringetin, which has a regiospecific OH-OMe substitution at positions 3’ and 5’ of myricetin. As we expected, syringetin significantly induced melanogenesis in B16F10 melanoma cells. The results indicated a crucial role for the 3’ and/or 5’-O-methyl groups in promoting melanin synthesis, despite the alterations in functionalities at other positions. 

This study aimed to investigate the impact of syringetin on melanin biosynthesis and its underlying molecular mechanisms in B16F10 mouse melanoma cells, with a focus on understanding the signaling pathways involved. The study found that syringetin exhibited no cytotoxic effects up to a concentration of 6 μg/mL. Moreover, it demonstrated a concentration-dependent increase in melanin content and intracellular tyrosinase activity in B16F10 cells. This led to the hypothesis that syringetin stimulates melanogenesis by modulating the melanogenic signaling pathway. To explore this further, the study utilized Western blot analysis to assess the expression of melanogenesis-related proteins in B16F10 cells treated with various doses of syringetin (ranging from 1.5 to 6 μg/mL) for 72 h.

In mammals, melanogenesis is governed by multiple enzymes, including tyrosinase, TRP-1, and TRP-2. Among the key regulators of melanogenesis in melanocytes is MITF, a major transcriptional factor responsible for controlling the expression of tyrosinase, TRP-1, and TRP-2. The study’s findings, presented in Figure 2, demonstrated that syringetin increased the protein expression of tyrosinase, TRP-1, TRP-2, and MITF.

Melanogenic factors are influenced by external stressors and are regulated through cross-regulation involving various signaling molecules. The cAMP signaling pathway has been reported as a regulator of melanogenesis, particularly through the cAMP/PKA/CREB pathway, which activates the expression of genes related to melanogenesis, including MITF, tyrosinase, TRP-1, and TRP-2 [17,18]. In this study, syringetin was found to significantly enhance PKA activity (Figure 3), suggesting that the melanogenic effect of syringetin is mediated via the cAMP/PKA/CREB signaling pathway. However, further validation is needed to ascertain the impact of syringetin on the cAMP/PKA signaling pathway by assessing cAMP production or CREB phosphorylation levels. To evaluate the impact of syringetin on the MAPK signaling pathway, the phosphorylation status of AKT was also assessed via Western blot analysis. As anticipated, syringetin inhibited AKT phosphorylation (Figure 4), providing conclusive evidence of syringetin’s role in downregulating melanogenesis through p-AKT modulation.

Mitogen-activated protein kinases (MAPKs), including p38 MAPK, extracellular signal-regulated kinase (ERK), and c-Jun N-terminal kinase (JNK), have been implicated in MITF expression regulation [19,20]. Additionally, several signaling pathways, such as ERK, p38, and phosphatidylinositol 3-kinase (PI3K)/AKT, have been linked to melanin synthesis regulation [21,22]. Recent studies have highlighted the ability of certain natural compounds to upregulate melanogenesis in B16F10 cells by downregulating P-ERK and upregulating p-38. Consequently, modulating the MAPK signaling pathway represents a potential approach to controlling melanogenesis. Results depicted in Figure 5 indicate that syringetin activates melanogenesis via the MAPK (ERK, JNK, and p-38) signaling pathway. 

By examining previous research, it was discovered that the interaction between MITF and β-catenin plays a crucial role in activating specific target genes regulated by MITF. Both MITF and β-catenin are essential mediators in the Wnt-signaling pathway, which is responsible for melanocyte differentiation. Notably, the inhibition of GSK3-mediated β-catenin phosphorylation is a significant event within the Wnt/β-catenin signaling pathway [23,24]. Therefore, to validate the influence of the GSK3β/β-catenin signaling pathway on melanogenesis stimulation by syringetin, we conducted Western blotting experiments. Figure 6 illustrates that the level of phospho-GSK3β also increased upon exposure to chrysoeriol, leading to the deactivation of GSK3β. Consequently, this resulted in an upregulation of β-catenin expression and a decline in phosphorylation.

In conclusion, our study provides evidence that syringetin induces melanin production and tyrosinase activity in B16F10 cells and sheds light on the underlying molecular mechanisms involved in melanogenesis. Our findings suggest that syringetin may regulate melanogenesis by stimulating the production of MITF, tyrosinase, TRP-1, and TRP-2. Understanding the molecular mechanisms of bioactive compounds and their specific targets is essential for their effective use. Our results show that the activation of the PKA/CREB, PI3K/AKT, MAPK, and Wnt/β-catenin signaling pathways by syringetin leads to the upregulation of MITF and subsequent activation of melanin production (Figure 7). After conducting a primary skin irritation test on human volunteers, it was concluded that the application of syringetin did not result in any adverse effects to the skin. As a result, our study suggests that syringetin has potential as a safe and effective agent for stimulating pigmentation, making it a promising candidate for use in cosmetics and in the medical treatment of hypopigmentation disorders. Although we have demonstrated the melanogenic effects of syringetin in vitro, further research is needed to determine its efficacy in vivo.

## 4. Materials and Methods

### 4.1. Materials

Syringetin (CAS 4423-37-4) was purchased from Extrasynthese (Genay Cedex, France), while α-melanocyte stimulating hormone (α-MSH), protease/phosphatase inhibitor cocktail, sodium hydroxide (NaOH), and L-DOPA were obtained from Sigma-Aldrich (St. Louis, MO, USA). MTT, DMSO, PBS, TBS, SDS, RIPA buffer, and the ECL kit were purchased from Biosesang (Seongnam, Gyeonggi-do, Korea), while DMEM, penicillin–streptomycin, BCA protein assay kits, and 0.5% trypsin-ethylenediaminetetraacetic acid (10×) were obtained from Thermo Fisher Scientific (Waltham, MA, USA). For Western blots, the primary antibodies against tyrosinase, TRP-1, TRP-2, and MITF were purchased from Santa Cruz Biotechnology (Dallas, TX, USA), while the other antibodies, against p-ERK, ERK, p-p38, p38, p-JNK, JNK, p-PKA, PKA, p-AKT, AKT, p-GSK-3β, GSK-3β, p-β-catenin, β-catenin, β-actin, anti-rabbit, and secondary antibodies, were purchased from Cell Signaling Technology (Danvers, MA, USA). Skimmed milk was purchased from BD Difco (Sparks, MD, USA), fetal bovine serum (FBS) from Merck Millipore (Burling, USA), and Tween 20 and 2 × Laemmli sample buffer were obtained from Bio-rad (Hercules, CA, USA).

### 4.2. Cell Culture 

Mouse melanoma B16F10 cells were purchased from ATCC: The Global Bioresource Center (Manassas, VA, USA). The cells were cultured in DMEM supplemented with 10% FBS and 1% penicillin−streptomycin under a saturated atmosphere of 5% CO_2_ at 37 °C until they reached 70–80% confluence, before use in further experiments.

### 4.3. Cell Viability

The cells were cultured in complete DMEM, supplemented with different concentrations of syringetin (ranging from 0 to 6 μg/mL), in 24-well plates at an initial density of 1.5 × 10^4^ cells/well for 72 h. Following this, the MTT solution (0.2 mg/mL) was added to each well and incubated for 3 h at 37 °C. The produced formazan was solubilized by adding 500 μL of DMSO to each well, and the absorbance was measured at 570 nm using a microplate reader (Biotek; Winooski, VT, USA).

### 4.4. Determination of Cellular Melanin Content

B16F10 melanoma cells were seeded into 60 mm cell culture dishes at a density of 8.0 × 10^4^ cells/dish and allowed to attach overnight. The cells were then treated with syringetin (0–6 μg/mL) or α-MSH (100 nM) for 72 h, with arbutin (200 μM) serving as a positive control. Following treatment, the cells were washed with 1 × PBS and lysed with RIPA buffer containing 1% protease inhibitor cocktail at 4 °C for 20 min. The lysate was centrifuged at –8 °C for 20 min at 15,000 rpm to obtain a pellet, which was then dissolved in 1 N NaOH supplemented with 10% DMSO at 90 °C for 10 min. Aliquots of the cell lysates were transferred to 96-well culture plates, and the melanin content was estimated by measuring the absorbance at 405 nm using a microplate reader (Biotek; Winooski, VT, USA). The melanin content was normalized to the total amount of cellular protein and expressed as a relative value compared to untreated control cells.

### 4.5. Evaluation of Cellular Tyrosinase Activity

B16F10 melanoma cells were seeded into 60 mm cell culture dishes at a density of 8.0 × 10^4^ cells/dish and allowed to attach overnight. The cells were then treated with syringetin (0–6 μg/mL) and α-MSH (100 nM) for 72 h, with arbutin (200 μM) used as a positive control. Following treatment, the cells were washed with 1 × PBS and lysed at 4 °C for 20 min. The lysate was centrifuged at −8 °C for 20 min at 15,000 rpm to obtain the supernatants. The protein content of each supernatant was determined using a BCA assay kit, and equal amounts of protein from each sample were mixed with 2 mg/mL of L-DOPA (80 μL) and incubated at 37 °C for 1 h. The absorbance of the formed dopachrome was measured using a microplate reader (Biotek; Winooski, VT, USA) at 490 nm.

### 4.6. Western Blotting

B16F10 melanoma cells were seeded at a density of 8.0 × 10^4^ cells/dish in 60 mm cell culture dishes and treated with syringetin (0–6 μg/mL) or α−MSH (100 nM) for 4–40 h, with arbutin (200 μM) as a positive control. After washing the cells with 1 × PBS, lysis buffer was added and incubated at 4 °C for 20 min, followed by centrifugation at 15,000 rpm and −8 °C to obtain the supernatants. Protein quantification of the supernatants was determined using a BCA assay kit. To prepare the loading sample, equal amounts of protein and 2 × Laemmli sample buffer were mixed in a 1:1 ratio and heated at 100 °C for 5 m. Equal protein concentrations were electrophoresed using SDS−polyacrylamide gel to separate proteins by size. The proteins were transferred to a PVDF membrane, which was blocked in 5% skimmed milk dissolved in TBS−T (Tris−buffered saline with 1% Tween 20) for 2 h. After washing the membrane with 1 × TBS−T, the primary antibody was added at a ratio of 1:2000 and incubated overnight at 4 °C. Following primary antibody washing, the horseradish peroxidase (HRP)-conjugated secondary antibodies (1:1000) were added to the membrane, which was then incubated with the respective primary antibodies for 2 h at room temperature. After antibody washing, protein band signals were visualized using ECL chemiluminescence reagents and Fusion Solo S (24 rue de Lamirault, 77090 Collégien, France), and the band images were analyzed using ImageJ software 1.45s (NIH, Bethesda, MD, USA).

### 4.7. Human Skin Irritation Test

This study was conducted in accordance with ethical principles with voluntary consent of the subjects after the ethical and scientific validity was reviewed by the Institutional Review Board of Dermapro Inc., based on the PCPC guidelines and the Declaration of Helsinki, with the aim of determining the presence of primary irritation to human skin.

More than 30 healthy men and women aged 20 to 60 years without skin diseases who met the inclusion criteria were selected, and the purpose and method of the study, as well as possible adverse events, were explained to the selected subjects. They completed the informed consent form before participation in the study. Thirty-two female subjects who met the inclusion criteria and did not meet any of the exclusion criteria participated in this study, but one subject dropped out (DSA 220 22 26; voluntary withdrawal), leaving a total of 31 subjects who participated in the entire study. The mean age of the subjects was 43.19 ± 5.97 years, with the oldest being 53 years old and the youngest being 29 years old. The chosen test site, the back, was cleaned with 70% ethanol and then 20 μL of the test material was applied for 24 h. The first evaluation was performed 20 min after removal and the second evaluation was performed 24 h later. The skin reaction results for each test substance were calculated according to the following formula:$$\mathrm{Response}=\frac{\sum \left(Grade\times No.of\;Responders\right)}{4(Maximum\;Grade)\times n(Total\;Subjects)}\times 100\times 1/2$$

### 4.8. Statistical Analysis

The experimental data are presented as the mean ± standard deviation (SD) of three independent experiments. Statistical significance was determined using the Student’s *t*-test, and the level of significance is indicated as follows: # *p* < 0.001 vs. the unstimulated control group; * *p* < 0.05, ** *p* < 0.01, and *** *p* < 0.001 vs. the α-MSH treatment group alone.

## Figures and Tables

**Figure 1 ijms-24-09960-f001:**
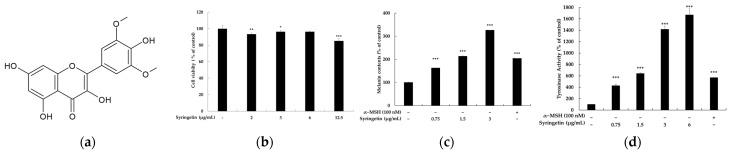
Molecular structure of syringetin (**a**) and its effects on melanogenesis (**b**–**d**). For the effect of syringetin on the viability of B16F10 melanoma cells (**b**), the cells were treated with syringetin (2, 3, 6, and 12.5 μg/mL) for 72 h, and the cytotoxicity of syringetin was evaluated using the MTT assay. For the effect of syringetin on melanin production (**c**) and tyrosinase activity (**d**) in B16F10 cells, the cells were treated with syringetin (0.75, 1.5, and 6 μg/mL) for 72 h, and α-MSH was used as the positive control. The results are presented as mean ± SD from three repeated experiments. * *p* < 0.05, ** *p* < 0.01, *** *p* < 0.001 vs. unstimulated control group.

**Figure 2 ijms-24-09960-f002:**
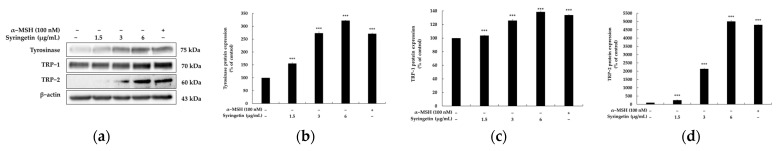
Effect of syringetin on tyrosinase, TRP−1, TRP−2, and MITF protein expression in B16F10 cells. The cells were treated with syringetin (1.5, 3, and 6 μg/mL) for 40 h. (**a**,**e**) Western blotting results, (**b**) tyrosinase protein expression, (**c**) TRP-1 protein expression, (**d**) TRP-2 protein expression, and (**f**) MITF protein expression. α-MSH was used as the positive control. The results are presented as mean ± SD from three repeated measurements using Image J. *** *p* < 0.001 vs. unstimulated control group.

**Figure 3 ijms-24-09960-f003:**
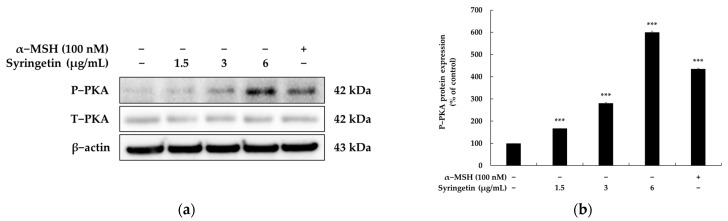
Effect of syringetin on PKA protein expression in B16F10 cells. The cells were treated with syringetin (1.5, 3, and 6 μg/mL) for 24 h. Western blot (**a**) and densitometric (**b**) analysis for P-PKA/T-PKA. α-MSH was used as the positive control. β-actin was used as a loading control. The results are presented as mean ± SD from three repeated measurements using Image J. *** *p* < 0.001 vs. unstimulated control group.

**Figure 4 ijms-24-09960-f004:**
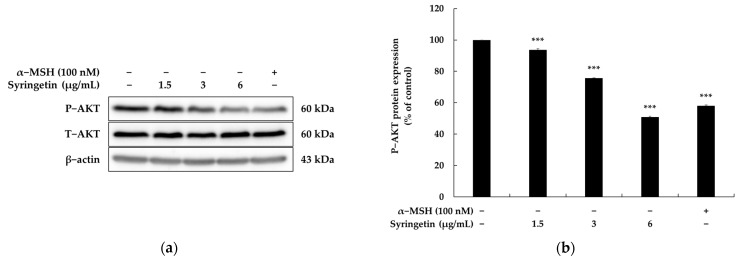
Effect of syringetin on AKT protein expression in B16F10 cells. The cells were treated with syringetin (1.5, 3, and 6 μg/mL) for 4 h. Western blot (**a**) and densitometric (**b**) analysis for p-Akt/Akt. α-MSH was used as the positive control. β-actin was used as a loading control. The results are presented as mean ± SD from three repeated measurements using Image J. *** *p* < 0.001 vs. unstimulated control group.

**Figure 5 ijms-24-09960-f005:**
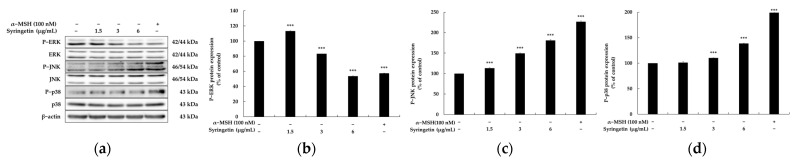
Effect of syringetin on MAPK protein expression in B16F10 cells. The cells were treated with syringetin (1.5, 3, and 6 μg/mL) for 4 h. (**a**) Western blotting results, (**b**) P-ERK protein expression, (**c**) P-JNK protein expression, and (**d**) P-p38 protein expression. α-MSH was used as the positive control. β-actin was used as a loading control. The results are presented as mean ± SD from three repeated measurements using Image J. *** *p* < 0.001 vs. unstimulated control group.

**Figure 6 ijms-24-09960-f006:**
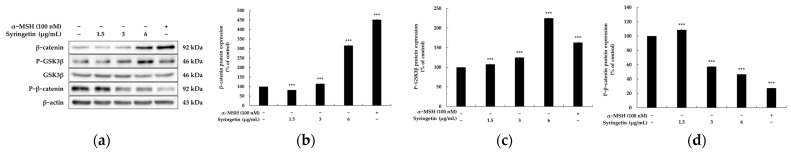
Effect of syringetin on β-catenin, P-GSK3β, and P-β-catenin protein expression in B16F10 cells. The cells were treated with syringetin (1.5, 3, and 6 μg/mL) for 24 h. (**a**) Western blotting results, (**b**) β-catenin protein expression, (**c**) P-GSK-3β protein expression, and (**d**) P-β-catenin protein expression. α-MSH was used as the positive control. β-actin was used as a loading control. The results are presented as mean ± SD from three repeated measurements using Image J. *** *p* < 0.001 vs. unstimulated control group.

**Figure 7 ijms-24-09960-f007:**
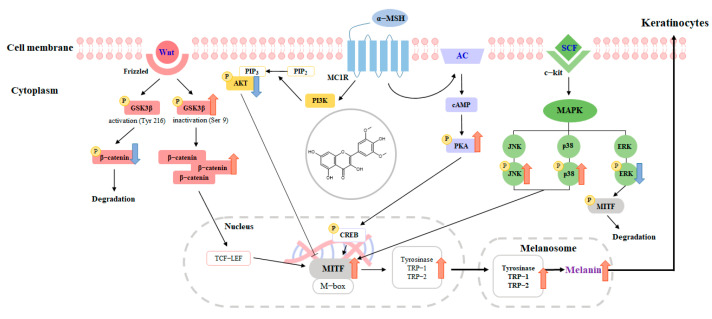
Schematic diagram of the proposed mechanism regulating the stimulative action of syringetin on melanogenesis.

**Table 1 ijms-24-09960-t001:** Results from the primary skin irritation tests on human volunteers (*n* = 31).

No.	Test Sample	No. of Respondents	20 min after Removal				24 h after Removal				Reaction Grade (R) *		
			+1	+2	+3	+4	+1	+2	+3	+4	24 h	48 h	Mean
1	Syringetin (3 μg/mL)	0	-	-	-	-	-	-	-	-	0	0	0
2	Syringetin (6 μg/mL)	0	-	-	-	-	-	-	-	-	0	1	0
3	Squalene	0	-	-	-	-	-	-	-	-	0	0	0

The investigator evaluated the reactions 20 min and 24 h after removing the treatment, following the PCPC guidelines (2014). * The range of irritation was classified as “no to slight irritation” with values ranging from 0.00 to less than 0.87.

## Data Availability

Not applicable.

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
