# Peer review of "Syringetin Promotes Melanogenesis in B16F10 Cells"

_ijms, 2023, doi:10.3390/ijms24129960_

Round 1

Reviewer 1 Report

It has been reported that Syringetin could stimuli Melanogenesis in B16F10 Cells.

Points of concern that need to be addressed by the authors are as follows:

1 The PI3K/Akt signaling pathway is claimed to be closely relevant to GSK3β in previous research, but PI3K-AKT-mTOR is a classic insulin signaling pathway. The author found key pathway response to Syringetin, why other protein in these pathway not selected? Thus, complementary functional recovery experiment is recommended, evaluate the key proteins or genes expression level in both upstream and downstream of PKA and PI3K/Akt signaling pathway. So the conclusion “The results suggested that syringetin upregulated the AKT/mTOR/MITF/tyrosinase pathway, leading to the expression of tyrosinase in B16F10 cells by suppressing the AKT/mTOR pathway.” should be caution in present manuscript.

2 Statistical analysis of western blot, the level of protein expression in the liver was examined and normalized using β-actin as an internal control. Please modify.

3 The cite of chart order confused. e.g. Figure 2(e) is upside down and not cited in manuscript; Fig.4, 5, 6 is not cited in manuscript. Please check all the charts carefully before submit.

4 The CAS No. of Syringetin should be provide.

 Moderate editing of English language

Author Response

<International Journal of Molecular Sciences>

< Syringetin Promotes Melanogenesis in B16F10 Cells>

Dear Editor,

Thank you for your useful comments and suggestions on the language and structure of our manuscript. We have modified the manuscript accordingly, and detailed corrections are listed below point by point. Our authors received English proofreading through native speakers before submitting paper (English-Editing-Certificate-65570, MDPI), and we will be done making corrections in English for the newly changed and inserted content paper (English-Editing-Certificate-66718, MDPI):

Reviewer #1

  1. The PI3K/Akt signaling pathway is claimed to be closely relevant to GSK3β in previous research, but PI3K-AKT-mTOR is a classic insulin signaling pathway. The author found key pathway response to Syringetin, why other protein in these pathway not selected? Thus, complementary functional recovery experiment is recommended, evaluate the key proteins or genes expression level in both upstream and downstream of PKA and PI3K/Akt signaling pathway. So the conclusion “The results suggested that syringetin upregulated the AKT/mTOR/MITF/tyrosinase pathway, leading to the expression of tyrosinase in B16F10 cells by suppressing the AKT/mTOR pathway.” should be caution in present manuscript.

→ We have removed the entry for "AKT/mTOR/MITF/tyrosinase pathway" as you pointed out. (Line 153-160)

  1. Statistical analysis of western blot, the level of protein expression in the liver was examined and normalized using β-actin as an internal control. Please modify.

→ We have inserted the statement "β-actin was used as a loading control." in all Western data descriptions as you pointed out.

  1. The cite of chart order confused. e.g. Figure 2(e) is upside down and not cited in manuscript; Fig.4, 5, 6 is not cited in manuscript. Please check all the charts carefully before submit.

We have corrected Figure 2(e) as you pointed out, and completed the references to Figures 4, 5, and 6.

  1. The CAS No. of Syringetin should be provide.

→ You have provided the CAS number of syringetin. (Line 338)

Reviewer 2 Report

Flavonoids are a group of polyphenolic compounds with a low molecular weight and are distributed widely as secondary metabolites in most plants. They exert mainly antioxidant and free radical scavenging effects, and have a variety of pharmacological and medicinal functions, including antiviral, anti-inflammatory, antiallergic and antitumor activities.

Many studies have focused on the utility of phenolic compounds as cosmetic materials.

Some studies have accumulated considerable evidence that flavonoids have different and even contradictory roles on melanin production according to the chemical structures.

The authors investigated whether Syringetin promotes melanogenesis. Their results indicated that Syringetin significantly promoted melanogenesis and tyrosinase activity in a dose-dependent manner. They also reported that Syringetin increased the expression of MITF, TRP-1 and DCT proteins. However, the experimental conditions used are already existing methods, and are not uncommon. Moreover, as I wrote in the Minor revision below, some points should be considered further and corrected errors. The reviewer's point of improvement is that the discussion is written together in the result, and the discussion is insufficient. Results and discussion should be described separately.

<Major revision>

Discussion is insufficient. For example, the following sentences should be described.

1) It is already known that flavonoid derivatives promote melanogenesis in human melanoma cells although there are some exceptions. The authors didn't mention well about this.

2) On the other hand, it should be known that Myricetin whose dimethoxy derivative is Syringetin has inhibited melanogenesis, which is related to the hydroxy groups of R1 and R3 of the B ring (Takekoshi et al., Tokai J. Exp. Clin. Med., 2014). In this connection the structure-activity relationship of Myricetin and Syringetin should be mentioned.

3) The saponins-rich fractions from argan leaves also inhibited melanin production in B16 murine melanoma cells (Villareal et al., Molecules, 2022).

4) More interestingly, it was suggest that Quercetin increases the tyrosinase activity at doses <10 μM, but strongly reduces cellular melanogenesis at more than 20 μM without toxicity in these cells (Yang et al., Phytother. Res. 2011). Based on this result, additional experiments with increasing concentrations of Syringetin should be performed.

5) The figure regarding molecular mechanisms of flavonoids on melanin synthesis should be depicted.

<Minor revisions>

1. Line 53: The pathway of DHI as well as the generation of DHICA by DCT should be described.

2. Line 56: Reference 12 is not appropriate. Boissy et al. (Experimental Dermatology, 7, 198–204) should be cited.

3. Line 75: “3,5,7,4′-tetrahydroxy-3′,5′dimethoxyflavone” should be read as “3,4′,5,7-tetrahydroxy-3′,5′-dimethoxyflavone”.

4. Line 124: Is correct the sentence that Syringetin successfully inhibits MITF expression in B16F10 cells?

5. Line 124: Fig. 2g is not depicted.

6. Fig. 2e: The image is inverted.

7. Fig.3a and 4a: A description of P-AKT and T-AKT should be given. An explanation using β-actin as a control is needed.

8. Line 179: Regarding the sentence that Syringetin treatment led to a significant increase in ERK phosphorylation, according to Figure 5b, it should be noted that P-ERK expression increased only at the administration of 1.5 ug/mL Syringetin and decreased thereafter.

9. Fig. 5 c and 5d do not correspond to P-JNK and P-p38 in Figure legend. Figures 5c and 5d are the same as Figures 2c and 2d.

10. Line 249: 10^4 should use superscript.

11. Line 348: Sci. Rep. is not bold and it should be used italic.

No comments.

Author Response

<International Journal of Molecular Sciences>

< Syringetin Promotes Melanogenesis in B16F10 Cells>

Dear Editor,

Thank you for your useful comments and suggestions on the language and structure of our manuscript. We have modified the manuscript accordingly, and detailed corrections are listed below point by point. Our authors received English proofreading through native speakers before submitting paper (English-Editing-Certificate-65570, MDPI), and we will be done making corrections in English for the newly changed and inserted content paper (English-Editing-Certificate-66718, MDPI):

Reviewer #2

  1. Flavonoids are a group of polyphenolic compounds with a low molecular weight and are distributed widely as secondary metabolites in most plants. They exert mainly antioxidant and free radical scavenging effects, and have a variety of pharmacological and medicinal functions, including antiviral, anti-inflammatory, antiallergic and antitumor activities. Many studies have focused on the utility of phenolic compounds as cosmetic materials. Some studies have accumulated considerable evidence that flavonoids have different and even contradictory roles on melanin production according to the chemical structures.

The authors investigated whether Syringetin promotes melanogenesis. Their results indicated that Syringetin significantly promoted melanogenesis and tyrosinase activity in a dose-dependent manner. They also reported that Syringetin increased the expression of MITF, TRP-1 and DCT proteins. However, the experimental conditions used are already existing methods, and are not uncommon. Moreover, as I wrote in the Minor revision below, some points should be considered further and corrected errors. The reviewer's point of improvement is that the discussion is written together in the result, and the discussion is insufficient. Results and discussion should be described separately.

→ We have rewritten the "Discussion" section with your suggestion. (Line 225-332)

  1. It is already known that flavonoid derivatives promote melanogenesis in human melanoma cells although there are some exceptions. The authors didn't mention well about this.

→ We have inserted information about flavonoids in the "Discussion" section as you pointed out. (Line 226-269).

  1. On the other hand, it should be known that Myricetin whose dimethoxy derivative is Syringetin has inhibited melanogenesis, which is related to the hydroxy groups of R1 and R3 of the B ring (Takekoshi et al., Tokai J. Exp. Clin. Med., 2014). In this connection the structure-activity relationship of Myricetin and Syringetin should be mentioned.

→ Conflicting results have been provided regarding the association between the hydroxy group of flavonoids and their structure and activity against melanogenesis. (Line 242-262)

  1. The saponins-rich fractions from argan leaves also inhibited melanin production in B16 murine melanoma cells (Villareal et al., Molecules, 2022).

→ The paper you provided reports on 18 natural products, including myricetin, quercetin, and gallic acid derivatives, and found that myricetin had no effect on melanogenesis in the HMV II cell line (Tokai J Exp Clin Med., Vol. 39, No. 3, pp. 116-121, 2014)

  1. More interestingly, it was suggest that Quercetin increases the tyrosinase activity at doses <10 μM, but strongly reduces cellular melanogenesis at more than 20 μM without toxicity in these cells (Yang et al., Phytother. Res. 2011). Based on this result, additional experiments with increasing concentrations of Syringetin should be performed.

→ There are four representative articles related to quercetin and melanogenesis.

(1) Phytother. Res. 2011. 25(8):1166-73 (2) Biosci. Biotechnol. Biochem. 2009. 73(9):1989-93 (3) J. Mol. Histol. 2004. 35(2):157-65 (4) Pigment Cell Res. 2004. 17(1):66-68 (5) Biosci. 2004. 17(1):66-73.

(1)In B16F20 cells, Quercetin increases the tyrosinase activity at doses <10 μM, but strongly reduces cellular melanogenesis at more than 20 μM.  (2) In B16F20 cells, quercetin inhibits melanogenesis in a concentration-dependent manner at concentrations of 10 Μm, 10 μM, and 10 μM. On the other hand, paper (3) shows that it inhibits melanogenesis in a three-dimensional reconstituted human epidermal culture model, and paper (4) shows that quercetin increases melanogenesis in human melanoma cells (HMVII).

In other words, the findings of the authors of paper (1), as you claim, are likely to be an experimental error that is not common.

  1. The figure regarding molecular mechanisms of flavonoids on melanin synthesis should be depicted.

→ A new picture of the molecular mechanism of flavonoids on melanin synthesis. (Line 333)

  1. Line 53: The pathway of DHI as well as the generation of DHICA by DCT should be described.

→ We described the path of DHI and the generation of DHICA by DCT as follows. “L-dopaquinone can be converted to cysteinyl-DOPA, which is further oxidized to pheomelanin, or spontaneously reacted to give, 5,6-dihydroxyindole (DHI) and 5,6-dihydroxyindole-2-carboxylic acid (DHICA), via dopachrome. The production of DHICA is accelerated by dopachrome tautomerase (also called tyrosinase-related pro-tein-2; TRP-2) or copper ions..

  1. Line 56: Reference 12 is not appropriate. Boissy et al. (Experimental Dermatology, 7, 198–204) should be cited.

→ We have fixed Reference 12 as you pointed out. (446-447)

  1. Line 75: “3,5,7,4′-tetrahydroxy-3′,5′dimethoxyflavone” should be read as “3,4′,5,7-tetrahydroxy-3′,5′-dimethoxyflavone”.

→ We have completed the fix as you pointed out. (Line 76)

  1. Line 124: Is correct the sentence that Syringetin successfully inhibits MITF expression in B16F10 cells?

→ We have completed the fix with "activates" as you pointed out. (Line 126)

  1. Line 124: Fig. 2g is not depicted.

→ We have completed the fix as you pointed out. (Line 141)

  1. Fig. 2e: The image is inverted.

→ We have completed the fix as you pointed out. (Line 141)

  1. Fig.3a and 4a: A description of P-AKT and T-AKT should be given. An explanation using β-actin as a control is needed.

→ We have completed the fix as you pointed out. (Line 148-152)

  1. Line 179: Regarding the sentence that Syringetin treatment led to a significant increase in ERK phosphorylation, according to Figure 5b, it should be noted that P-ERK expression increased only at the administration of 1.5 ug/mL Syringetin and decreased thereafter.

→ We have corrected that it only decreased at 3 μg/mL and 6 μg/mL as you pointed out. (Line 177-178)

  1. Fig. 5 c and 5d do not correspond to P-JNK and P-p38 in Figure legend. Figures 5c and 5d are the same as Figures 2c and 2d.

→ We have modified the picture as you pointed out.

  1. Line 249: 10^4 should use superscript.

→ We have completed the fix as you pointed out. (Line 361)

  1. . Line 348: Sci. Rep. is not bold and it should be used italic.

→ We have completed the fix as you pointed out. (Line 446)

Round 2

Reviewer 1 Report

The authors did not supplement the experiment, but modified it as suggested, therefore recommend accepted.

Minor editing of English language required

Reviewer 2 Report

The author adequately answered all the reviewer's questions. The reviewer recommends the acceptance of this form.